# Endothelial-Derived Extracellular Vesicles Induce Cerebrovascular Dysfunction in Inflammation

**DOI:** 10.3390/pharmaceutics13091525

**Published:** 2021-09-21

**Authors:** David Roig-Carles, Eduard Willms, Ruud D. Fontijn, Sarai Martinez-Pacheco, Imre Mäger, Helga E. de Vries, Mark Hirst, Basil Sharrack, David K. Male, Cheryl A. Hawkes, Ignacio A. Romero

**Affiliations:** 1School of Life, Health and Chemical Sciences, Biomedical Research Network, Open University, Milton Keynes MK7 6AA, UK; david.roig.carles@gmail.com (D.R.-C.); saraimartinezpacheco@gmail.com (S.M.-P.); mark.hirst@open.ac.uk (M.H.); david.male@open.ac.uk (D.K.M.); 2Department of Biochemistry and Genetics, La Trobe Institute for Molecular Science, La Trobe University, Bundoora, VIC 3086, Australia; E.Willms@latrobe.edu.au; 3Amsterdam UMC, Department of Molecular Cell Biology and Immunology, MS Centre Amsterdam, de Boelelaan 1117 VU University, 1081 HZ Amsterdam, The Netherlands; r.fontijn@amsterdamumc.nl (R.D.F.); he.devries@vumc.nl (H.E.d.V.); 4Department of Paediatrics, University of Oxford, Oxford OX3 9DU, UK; imre.mager@paediatrics.ox.ac.uk; 5Academic Department of Neuroscience and Sheffield NIHR Translational Neuroscience BRC, Sheffield Teaching Hospitals, NHS Foundation Trust, University of Sheffield, Sheffield S10 2JF, UK; b.sharrack@sheffield.ac.uk; 6Department of Biomedical and Life Sciences, Lancaster University, Lancaster LA1 4YW, UK; c.hawkes@lancaster.ac.uk

**Keywords:** blood–brain barrier, cell-to-cell communication, exosomes, extracellular vesicles, neuroinflammation

## Abstract

Blood–brain barrier (BBB) dysfunction is a key hallmark in the pathology of many neuroinflammatory disorders. Extracellular vesicles (EVs) are lipid membrane-enclosed carriers of molecular cargo that are involved in cell-to-cell communication. Circulating endothelial EVs are increased in the plasma of patients with neurological disorders, and immune cell-derived EVs are known to modulate cerebrovascular functions. However, little is known about whether brain endothelial cell (BEC)-derived EVs themselves contribute to BBB dysfunction. Human cerebral microvascular cells (hCMEC/D3) were treated with TNFα and IFNy, and the EVs were isolated and characterised. The effect of EVs on BBB transendothelial resistance (TEER) and leukocyte adhesion in hCMEC/D3 cells was measured by electric substrate cell-substrate impedance sensing and the flow-based T-cell adhesion assay. EV-induced molecular changes in recipient hCMEC/D3 cells were analysed by RT-qPCR and Western blotting. A stimulation of naïve hCMEC/D3 cells with small EVs (sEVs) reduced the TEER and increased the shear-resistant T-cell adhesion. The levels of *microRNA-155,* VCAM1 and ICAM1 were increased in sEV-treated hCMEC/D3 cells. Blocking the expression of VCAM1, but not of ICAM1, prevented sEV-mediated T-cell adhesion to brain endothelia. These results suggest that sEVs derived from inflamed BECs promote cerebrovascular dysfunction. These findings may provide new insights into the mechanisms involving neuroinflammatory disorders.

## 1. Introduction

BECs are the main cellular component of the BBB, a specialised feature of the vasculature of the central nervous system (CNS) [1]. BEC dysfunction is common in neuroinflammatory pathologies like multiple sclerosis (MS) [2]. Inflammatory modulators such as proinflammatory cytokines are upregulated in the plasma of individuals with MS and other conditions such as sepsis [3,4]. Cytokines play an additional central role in the progression of inflammation by acting on the BBB. For instance, tumour necrosis factor alpha (TNFα) and interferon gamma (IFNy) modulate the vascular function by increasing the paracellular permeability [5] and leukocyte adhesion to BECs and migration across BECs [6,7]. BECs express junctional complexes that contain tight junctional (TJs) and *adherens* junctional (AJs) proteins that are linked to the cytoskeleton via scaffolding proteins such as zona occludens-1 (ZO-1) and which confer the barrier properties of the BBB [8]. The expression and location of these complexes are altered in the vasculature in MS lesions [9] and in cytokine-treated BECs [10]. 

In addition to cytokines, vascular functions are also modulated by EVs [11]. EVs are lipid membrane-enclosed carriers of many different molecules, including proteins and both coding and non-coding RNAs (e.g., microRNAs (miRNAs)) that can influence the protein expression and the function of recipient cells [12]. EVs represent a heterogeneous population, but they are typically classified into apoptotic bodies, microvesicles and exosomes [13]. Apoptotic bodies are larger vesicles (500–5000 nm) that are produced as a consequence of cells undergoing apoptosis [12]. Microvesicles (100–1000 nm) are generated by the direct outward budding of the plasma membrane, whereas exosomes (30–200 nm) are produced intracellularly after the fusion of the multivesicular body with the plasma membrane and are released into the extracellular space [14]. Recent reports have highlighted the heterogeneous nature of EVs and the existence of additional EV subpopulations [15,16]. Due to the lack of optimal methods to isolate and identify specific EV populations, the current International Society of Extracellular Vesicles (ISEV) guidelines [12] recommends the classification of EVs as either small EVs (sEVs) (<200 nm) or large EVs (L-EVs) (200 nm–1000 nm), unless the specific EV origin is known. 

The precise role of EVs in the pathogenesis of MS is still unclear, but the number of circulating EVs has been shown to be increased in the plasma of patients with neuroinflammatory conditions [17]. Marcos-Ramiro and colleagues observed an increase in the number of endothelial-derived EVs in the plasma of MS patients compared to healthy controls [18]. They also observed that plasma-derived EVs reduced the expression of TJ protein zona-occludens (ZO-1) at the junctional area and decreased the electrical resistance of BECs. Yamamoto et al. suggested that EVs containing inflammation-induced miRNAs that were isolated from mouse BECs modulated the transcriptome of recipient pericytes [19]. In another study, EVs derived from human brain endothelial cells after treatment with the proinflammatory cytokine TNFα were found to carry higher levels of proinflammatory proteins (e.g., intercellular adhesion molecules 1 (ICAM1) and vascular cell adhesion molecule 1 (VCAM1)) compared to EVs isolated from naïve cells [20].

The role of BEC-derived EVs in cerebrovascular function generally, and in the context of inflammation specifically, is still poorly understood. Therefore, the aim of this study was to investigate the role of EVs derived from cytokine-challenged BECs (cytokine-EVs) in cerebrovascular functions. We found that the cytokine challenge of hCMEC/D3 cells induced an increased release of sEVs that contained proinflammatory modulators. The incubation of naïve BECs with these sEVs decreased the tightness of the endothelial monolayer and increased the T-cell adhesion to the endothelium. These findings suggest that the damage to the BBB during a neuroinflammatory insult may result in part from a feed-forward mechanism in which the uptake of EVs secreted from inflamed BECs by naïve cells induces a proinflammatory environment in the recipient BECs. These results may have implications for the pathophysiology of neuroinflammatory conditions in the brain vasculature. 

## 2. Materials and Methods

### 2.1. Cell Culture

The hCMEC/D3 cell line was cultured in endothelial cell basal medium-2 (EBM-2) supplemented with 0.025% (*v*/*v*) recombinant human epithelial growth factor (rhEGF), 0.025% (*v*/*v*) vascular endothelial growth factor (VEGF), 0.025% (*v*/*v*) insulin growth factor (IGF), 0.1% (*v*/*v*) recombinant human fibroblast growth factor (rhFGF), 0.1% (*v*/*v*) gentamycin, 0.1% (*v*/*v*) ascorbic acid, 0.04% (*v*/*v*) hydrocortisone and 2.5% (*v*/*v*) foetal bovine serum (FBS) (Lonza, Wilford, Nottinghamshire, UK), hereafter referred to as the endothelial complete medium. hCMEC/D3 cells were used from passages 25–35 and grown in calf skin collagen-coated tissue culture flasks until confluence unless otherwise indicated. The T-cell line Jurkat from acute T cells was a kind gift from Dr V Male (Cambridge University, UK). Jurkat cells were grown in suspension in RPMI 1640 with GLUTAMAX I (Gibco^®^Invitrogen, Paisley, Renfrewshire, UK) culture medium containing 10% (*v*/*v*) FBS and 100-μg/mL streptomycin + 100-units/mL penicillin (P/S). All cell lines were maintained in a 95% humidified air and 5% CO_2_ incubator at 37 °C. The cells were routinely screened for the presence of mycoplasma using the MycoAlert Mycoplasma kit (Lonza, Wilford, Nottinghamshire, UK).

### 2.2. Isolation of Brain Endothelial Cell-Derived Extracellular Vesicles

#### 2.2.1. Conditioned Cell Media Preparation

The isolation of BEC-derived EVs was carried out using hCMEC/D3 cells cultured in collagen-coated 175-cm^2^ tissue culture flasks for 48 h. Then, the cells underwent three Hanks Balanced Salt Solution (HBSS) washes before the cell culture media was replaced by 15 mL of endothelial complete medium supplemented with 2.5% EV-depleted FBS (hereafter referred to as sEV medium) (Gibco^®^Invitrogen, Paisley, UK). A combination of 10 ng/mL of both TNFa and IFNy diluted in cell sEV medium was used to treat hCMEC/D3 cells. Cells incubated with only cell sEV medium were used as the control. The cell-conditioned medium (CCM) was collected 24 h after the media exchange when the cells reached 95–100% confluency (approximately 15 × 10^6^ cells per 175-cm^2^ tissue culture flask). To remove living/dead cells, CCM was centrifuged at 3″× *g* for 10 min (min). Then, the cell debris was depleted by centrifugating CCM at 2000× *g* for 20 min. 

#### 2.2.2. Isolation of Large Extracellular Vesicles

To isolate L-EVs, CCM was centrifuged at 10,000× *g* for 30 min, followed by a wash using 30 mL of 1× phosphate-buffered saline (PBS). Resuspended L-EVs were filtered by a 0.8-µm sterile filtered unit and centrifuged at 10,000× *g* for 30 min. The L-EVs were stored at −80 °C for up to two months to minimise the degradation of the RNA and vesicle number [21].

#### 2.2.3. Isolation of Small Extracellular Vesicles

Isolation of the sEVs was perform on the supernatant from the first L-EV centrifugation. Due to the small size of sEVs, ultracentrifugation at 120,000× *g* for 120 min was performed on CCM and followed by resuspending the pellet in 30 mL of PBS and repeating the 120,000× *g* spin for 120 min. Pelleted sEVs were resuspended in the appropriate buffer (PBS, endothelial sEV medium lacking IGF, EGF, VEGF and FBS or RNA/protein lysis buffer). The aliquots were stored at −80 °C for up to two months. A Rhe Sorvall Discovery (Brea, CA, USA) Superspin 630 Sorvall rotor (k factor 226.3) was used to perform all the ultracentrifugation steps, which were carried out at 4 °C. 

##### Endothelial-Derived Extracellular Vesicles Characterisation

A NS500 nanoparticle analyser (NanoSight, Malvern, Worcestershire, UK) was used for the nanoparticle tracking analysis (NTA). The particles appeared as sharp individual dots after adjusting the camera focus. Three 30-s videos were recorded for each sample, with a delay of 5 s between each recording. 

For transmission electron microscopy (TEM) of the EV samples, 10 µL of either L-EVs or sEVs was mixed 1:1 with 4% (*w*/*v*) paraformaldehyde (PFA). A 5-µL drop of this mix was incubated in a Formvar carbon-coated grid for 20 min at room temperature (RT). The grids were washed in PBS and incubated with 2.5% (*v*/*v*) glutaraldehyde for 5 min at RT. Glutaraldehyde was washed in distilled water for 2 min seven times. The grids were then counter-stained with 3.5% (*w*/*v*) of uranyl acetate for 20 s. The grids were air-dried and were observed under TEM (JEM1400) at 80 kV.

##### TNFα and IFNγ ELISA

An enzyme-linked immunosorbent assay (ELISA) from the DuoSet^®^ kit (R&D Systems, Abingdon, Oxfordshire, UK) was used to measure the residual levels of TNFα and IFNγ in sEVs following the manufacturer’s instructions. Briefly, 96-well plates were coated with either anti-TNFα or anti-IFNγ capture antibodies, washed and blocked in 0.1% (*w*/*v*) bovine serum albumin (BSA). The cytokine standards and sEVs (10^8^ sEVs/µL) were loaded onto the plates, incubated for 2 h at RT, then washed and the secondary antibody was added for one extra hour at RT. Then, the wash step was repeated, followed by the addition of the substrate solution. The plates were incubated for 20 min before the reaction was stopped. The signal was measured using a FLUOstar Optima fluorescence plate reader (BMG LABTECH, Aylesbury, UK) at 450 nm, and the background was subtracted by measuring the wavelength at 570 nm.

##### Uptake of Endothelial-Derived Small EVs

For the uptake experiments, the sEVs were prepared following Section 2.2. However, after the first 120,000× *g* centrifugations, the sEVs were labelled with 10% (*v*/*v*) Vybrant-DiO (Life technology, Paisley, UK) diluted in 5 mL of PBS for 30 min at 37 °C. Vybrant-DiO was added in the same ratio to PBS, and the sample was treated in the same manner and used as a control for any free Vybrant-DiO dye left in the solution. 

BECs were grown in 12-well plates to sub-confluence (95%) and incubated with DiO-labelled sEVs at the concentrations (from 0.1 to 10^8^ sEVs/µL) and times (from 6 to 48 h) specified in each figure. Then, the washed and tripsised cells were analysed on the FACs Calibur (Becton-Dickinson, Berkshire, Reading, UK) (FL1 detector set at 530 V). The median fluorescence of 10,000 cell events were reported. Collagen- and fibronectin- coated Nunc^®^ Labteck chamber slides were used to grow hCMEC/D3, as described elsewhere [22]. Then, 0.5 × 10^8^ cells/µL DiO-labelled sEVs were incubated with hCMEC/D3 cells for 6, 24 and 48 h. Then, the cells were fixed in 4% (*w*/*v*) PFA for 10 min at RT. The slides were mounted with mounting media containing DAPI dye (Vector Laboratories, Burlingame, CA, USA) for nuclear staining. The acquisition and analysis of the images were performed using confocal laser scanning microscopy (Leica TCS SP5, Leica Microsystems, Milton Keynes, UK). The images were shown as the maximum projection of a z-stack of images.

##### Electric Cell-Substrate Impedance Sensing (ECIS)

ECIS Z-Theta (Applied BioPhysics, Troy, NY, USA) was used to measure the TEER of the hCMEC/D3 cell monolayer in real time, as described previously (Keese et al., 2004). Twenty-thousand cells were seeded onto each well of a 96W10E+ array previously coated with calf skin collagen type I and grown until confluence. The cells were treated with sEVs at the doses (from 0.1 to 10^8^ sEVs/µL) described at the figure legends. A small amount (1 ng/mL) of TNFα and IFNγ was used as the positive control to induce the loss of endothelial resistance. Impedance data were collected at multiple frequencies and mathematically modelled to calculate the resistivity of cell–cell contacts Rb (Ω × cm^2^) at each time point measured [23]. Rb data are shown as a percentage of the control at each time point. 

##### Paracellular Permeability Assay

The effect of cytokine-sEVs on hCMEC/D3 cells was studied following a paracellular permeability assay previously described elsewhere [24,25]. Briefly, confluent hCMEC/D3 cells were incubated with 0.1 cytokine-sEVs/µL for 6, 24 and 48 h. Then, the apical media was replaced by 2 mg/mL of 70-KDa FITC-dextran in phenol red-free DMEM. The fluorescent signal in the basolateral chamber was read every 5 min during a 30-min window using a BMG plate reader and the derived permeability coefficient P_e_ [24].

##### Leukocyte Adhesion to Endothelium under Flow Conditions

T-cell adhesion was measured using a flow-based adhesion assay adapted from previously published data [26,27]. Briefly, hCMEC/D3 cells were grown in Ibidi^®^ μ-Slide VI0.4 (Ibidi^®^ GmbH, Martinstreid, Germany) until they reached confluence, then incubated with sEVs at the doses (from 0.1 to 10^8^ sEVs/µL) and times (from 6 to 48 h) described in their specific figure legends in static conditions and washed before the flow adhesion assay. The positive control consisted of 1-ng/mL TNFα and IFNγ incubation for 24 h [28]. To address the functions of VCAM1 and ICAM1 in this study, 30 µg/mL of neutralising antibodies against ICAM1, VCAM1 or the control mouse IgG (R&D Systems, Abingdon, Oxfordshire, UK) were added to washed hCMEC/D3 cells and incubated for 1 h at 37 °C in EBM-2 basal media. Some (2 × 10^6^ cells/mL) Jurkat T cells were labelled with 5-µM 5–chloromethylfluoresceindiacetate (CMFDA, Life Technologies, Eugene, OR, USA). The cells flowed through the channel containing endothelial monolayers at 0.5 dyn/cm^2^ for 5 min. Subsequently, the flow was increased to 1.5 dyn/cm^2^ for 1 min to remove non-adhered leukocytes. Interactions between the leukocyte and endothelial cells were recorded for 6 min; following which, the leukocyte adhesion was quantified. Five to ten different fields of vision (FOVs) along the centre of the channel were imaged using an inverted fluorescence microscope (Olympus IX70, Tokyo, Japan) with a ×10 objective. The firmly adhered leukocytes were counted manually. 

##### RT-qPCR 

The RNA was isolated from either cultured cells or the sEV pellet using a miRCURY Exiqon kit (Qiagen, Manchester, Greater Manchester, UK) following manufacturer’s instructions. Briefly, the cells or sEV pellet were incubated with the lysis buffer, homogenised by vortexing and precipitated with pure ethanol. The samples were cleaned and concentrated using miRCURY columns, washed and eluted in 50 µL of elution buffer. The RNA concentration and quality was measured using Nanodrop ONE (Life technology, Paisley, Renfrewshire, UK).

To investigate the effects of the cytokines and sEVs on the transcriptomes of BECs, hCMEC/D3 cells were grown to confluence and treated with 10^8^ sEVs/µL or 10 ng/mL of cytokines (TNFα and IFNy) for 24 h. The RNA was isolated as described above. Reverse transcription for microRNAs was performed using a Taqman microRNA transcription kit with specific Taqman primers for *miRNA-155-5p, 126-5p, 126-3p, 24, 146a* and *146b* following the manufacturer’s instructions. Small nuclear RNA *U6* (U6) was used as the control in BECs, whereas *let-7g* was used as the control of the microRNAs levels in sEVs [29]. For the mRNA analysis, total cDNA was generated using a TaqMan High-Capacity cDNA Reverse Transcription kit (Applied Biosystem, Life Technologies, Warrington, Lancashire, UK) using random primers. Quantitect SyberGreen master mix (Qiagen, Manchester, Greater Manchester, UK) was used to study the relative levels of the mRNAs with 10 ng of cDNA. Primers for *ICAM1, VCAM1* and Carcinoembryonic antigen-related cell adhesion molecule 1 (*CEACAM1*) were tested, and *β-actin* was used as a housekeeping gene (Merk Millipore, Watford, Hertfordshire, UK).

The relative amounts of microRNA and mRNA were calculated using the 2−ΔΔCt (delta-delta Ct) method [30] and normalised with the appropriate available internal control. The relative levels of microRNA/mRNA in treated hCMEC/D3 cells or cytokine-sEVs were expressed as fold changes over levels in unstimulated hCMEC/D3 cells or quiescent sEVs, respectively.

##### Western Blotting

Isolated cytokine-sEVs or hCMEC/D3 cells were resuspended in 1× RIPA Buffer (20-mM Tris-HCl (pH 8.0), 150-mM NaCl, 1-mM EDTA, 0.1% SDS, 1% Igepal, 50-mM NaF and 1-mM NaVO_3_) supplemented with protease inhibitor cocktail (Merk Millipore, Watford, Hertfordshire, UK). The samples were sonicated (Fisherbrand™ Model 120 Sonic Dismembrator) at 20% amplitude for 10 s on ice. The protein concentration was measured using a DC™ Protein Assay kit (Bio-Rad Laboratories Ltd., Watford, Hertfordshire UK). The samples were mixed with 2× Tris glycine sample buffer (Life Technologies, Paisley, Renfrewshire, UK), and the proteins were denatured by heating the sample at 75 °C for 5 min. Five micrograms of cytokine-sEV sample or 20 µg of cell lysate were loaded in 4–20% Tris-Glycine gels (Life technology, UK) and run at 120 V for 2 h. The proteins were transferred to a nitrocellulose membrane, and the membrane was blocked with 8% (*w*/*v*) skimmed milk diluted in Tris buffer saline containing 0.2% *w*/*v* Tween-20 (TBS-T). The membranes were incubated overnight at 4 °C with primary antibodies diluted in 8% (*w*/*v*) skimmed milk diluted in TBS-T. For sEV characterisation, the membranes were incubated with rabbit anti-mouse primary antibodies against CD9, CD63 or HSP70 (System Bioscience Palo Alto, CA, USA) at 1:1000. Membranes containing hCMEC/D3 lysates were incubated with mouse anti-human VCAM1 (1:300) and ICAM1 (1:300) (R&D Systems, Abingdon, Oxfordshire, UK), rabbit anti-human occludin (1:150) and claudin-5 (1:150) (Life technology, UK). Then, three 10-min washes with TSB-T were applied to the membranes, followed by an incubation with the appropriate secondary antibody, goat anti-rabbit or anti-mouse HRP (System Bioscience Palo Alto, CA, USA) at 1:5000 in 8% (*w*/*v*) of skimmed milk diluted in TBS-T for 60 min at RT. Subsequently, the membranes were washed 6 times with TBS-T for 10 min, followed by incubation with an enhanced chemiluminescence reagent (ECL) (GE Healthcare, Little Chalfont, Buckinghamshire, UK). The membranes were visualised using G:Box (Syngene, Cambridge, Cambridgeshire, UK)

##### Statistical Analysis

All data were presented as the mean ± SD (standard deviation). The number of independent experiments (*n*) with replicates was specified in each legend. The normality was assessed with the Shapiro–Wilk test, *p* = 0.05. The means were compared using unpaired or paired two-tailed *t*-tests for single comparisons and one-way or two-way ANOVA for multiple comparisons. The ANOVA analysis was followed by Dunnett’s or Tukey’s multiple comparisons tests. The specific analysis was specified in each figure legend. The normality of the data was assessed by the Shapiro–Wilk test. All tests were performed using the statistical software GraphPad Prism 7 (GraphPad Software, San Diego, CA, USA). *p* < 0.05 was considered to be statistically significant. 

## 3. Results

### 3.1. hCMEC/D3 Cells Secrete More Small EVs than Large EVs after Treatment with TNFα and IFNγ

To determine the effect of an inflammatory challenge on EV secretion, the number, size and morphology of EVs isolated from naïve BECs and BECs treated with proinflammatory cytokines (Figure 1a) was characterised using NTA, TEM and Western blotting. The NTA analysis revealed that the stimulation with proinflammatory cytokines increased the number of both cytokine-sEVs and a trend for number of cytokine-L-EVs (Figure 1b). The size distribution showed heterogeneous populations of sEVs and L-EVs (Figure 1c,e). The cytokine treatment had no effect on the mode of the diameter of both sEVs and L-EVs (Figure 1c). However, the size range of cytokine-L-EVs measured by D90 was greater than the rest of the isolated EV groups (Figure 1d). D10, D50 and D90 values indicate the percentage of particles (10, 50 and 90%, respectively) less than or equal to the corresponding particle size [31] (Figure 1e).

TEM revealed lipid-based structures with a spherical shape for larger vesicles (>200 nm) and a mix of spherical and cup-shaped morphology for smaller vesicles (<200 nm) (Figure 1f). As previously reported (Dozio and Sanchez 2017), cytokine-sEVs expressed markers of EVs, including CD9, CD63 and HSP70, which were also expressed in the cell lysate (Figure 1g). 

To evaluate how efficiently the method of EV isolation separated soluble cytokines from cytokine-sEVs, conditioned cell media (CCM) and sEVs were assessed for concentrations of TNFα and IFNγ. The concentration of residual TNFα and IFNy in the CCM following a 24-h cytokine incubation with BECs was 1.95 ± 0.52 and 2.37 ± 0.94 ng/mL, respectively. In isolated EVs, concentrations of TNFα and IFNy were 0.012 ± 0.014 ng/mL and 0.015 ± 0.008 ng/mL, respectively (Figure 1h), suggesting that less than 1% of residual TNFα and IFNγ was contained within the EV mixture. 

### 3.2. Uptake of Small EVs by Brain Endothelium Is Time- and Dose-Dependent

To determine the kinetics of the uptake of cytokine-sEVs by BECs, cytokine-sEVs were labelled with the lipophilic dye DiO, and their uptake by hCMEC/D3 cells was analysed using confocal microscopy and flow cytometry. Confocal microscopy showed a predominately perinuclear localisation of DiO staining with some staining dispersed in the cytoplasm (Figure 2a). The EV uptake was qualitatively similar at the 24-h and 48-h time points, suggesting that a level of saturation was achieved by 24 h. Based on these observations, we chose the 24-h end point for the dose response experiments using flow cytometry. A flow cytometry analysis of the mean fluorescence of DiO-labelled cytokine-sEVs revealed that the uptake of cytokine-sEVs by hCMEC/D3 cells was also dose-dependent (Figure 2b). 

### 3.3. Cytokine-Derived Small EVS Modulate the Transendothelial Resistance

The loss of paracellular barrier function is an established characteristic of brain endothelial cell dysfunction, which can be studied by measuring the electrical impedance of the endothelial monolayer [32]. Thus, we evaluated the effect of cytokine-sEVs on the impedance of BECs. A dose-dependent decrease of the TEER values was observed after the incubation of hCMEC/D3 cells with cytokine-sEVs (0.1, 0.5 and 1 × 10^8^ sEVs/µL) (Figure 3a). We observed a peak in decreased TEER at 83 ± 6% in comparison to the control cells with 1 × 10^8^ cytokine-sEVs/µL. However, longer incubation times did not cause a further decrease in TEER (Figure 3a). Next, hCMEC/D3 cells were treated with either quiescent BECs or cytokine-treated BECs (0.5 × 10^8^ sEVs/µL) to further characterise the specific effect of cytokine-sEVs on TEER. We observed that, after 10 h of incubation with sEVs, a decreased endothelial resistance was only observed in sEVs derived from cytokine-activated hCMEC/D3 cells (Figure 3b). No significant effect was observed with quiescent sEVs. To further confirm the role of cytokine-sEVs modulating the BBB permeability, the para-cellular permeability of 4-KDa FITC-Dextran was evaluated (Figure 3c). Incubation with 0.1 × 10^8^ cytokine-sEVs/uL showed a small but significant increase in para-cellular permeability after 6 h of treatment. This effect was transient, since a longer incubation only showed a small trend for increased para-cellular permeability.

### 3.4. Adhesion of T Cells to Brain Endothelium Is Modulated by Cytokine-sEVs

T-cell adhesion to BECs is a main feature of endothelial dysfunction in neuroinflammation [33]. Therefore, we decided to investigate the effect of cytokine-sEVs on T-cell adhesion to brain endothelia. For this, we used a flow-based assay in order to expose the cells to shear stress, which has been demonstrated to more closely mimic in vivo conditions [34]. The interaction of Jurkat T cells with hCMEC/D3 cells was studied using time-lapse microscopy (Figure 4a). A concentration-dependent increase in adhesion was observed, with a peak at 1 × 10^8^ cytokine-sEV/µL of 96 ± 23 firmly adhered Jurkat T cells per FOV (Figure 4b). In accordance with the results from the TEER experiments, a rapid increase of T-cell adhesion to BECs was observed after 6 h of incubation and was stable at later time points (Figure 4c). The T-cell adhesion after incubation with cytokine-sEVs was approximately half that of the number of cells that adhered after the incubation of BECs with 1 ng/mL of TNFα and IFNy (Figure 4b). Quiescent sEVs had no effect on the T-cell adhesion to naïve brain endothelium (Figure 4d).

### 3.5. Levels of Proinflammatory miRNA-155-5p and Adhesion Molecule mRNAs Are Increased in the Cargo of Cytokine-sEVs

Since EVs are known to carry genetic material that may influence BEC functions, we next decided to investigate the differences in the miRNA contents of cytokine-sEVs compared to quiescent sEVs. We first selected a series of miRNAs that are known to be upregulated (*miRNA-155-5p, miRNA-146a* and *miRNA-146b*); downregulated (*miRNA-126-5p* and *miRNA-126-3p*) or unchanged (*miRNA-24*) in hCMEC/D3 cells after treatment with TNFα and IFNy and whose role in modulating hCMEC/D3 functions is known [35]. All miRNAs were detectable in both cytokine- and quiescent sEVs (Figure 5a). The RT-qPCR analysis of the vesicle cargo revealed that the *miRNA-155-5p* expression was increased 8.20 ± 2.10-fold (*p* = 0.004) in cytokine-sEVs compared to quiescent EVs (Figure 5a). The anti-inflammatory *miRNA-126-3p* showed a trend towards downregulation in cytokine-sEVs (0.75 ± 0.56-fold change, *p* = 0.49), but this was not statistically significant (Figure 5a). No difference was observed in the expression of *miRNA-146a, -146b, 126-5p* and *-24* between quiescent and cytokine-sEVs (Figure 5a). The levels of *miRNA-155-5p* in cytokine-sEVs were confirmed to correspond with the endogenous expression of this miRNA in BECs after stimulation with 10 ng/mL of TNFα and IFNy for 24 h (6.40 ± 0.98-fold change, *p* = 0.0007) (Figure 5b). We also investigated the relative levels of mRNAs that are directly involved in T-cell adhesion and observed that *ICAM1* mRNA was significantly increased in cytokine-sEVs (6.21 ± 2.29, *p* = 0.017), whereas *VCAM1* mRNA showed a strong trend (3.77 ± 1.62, *p* = 0.054) (Figure 5c). Similar to previous reports [22], the upregulation of *ICAM1* and *VCAM1* mRNAs was also observed in hCMEC/D3 cells upon cytokine treatment (Figure 5d). By contrast, the mRNA levels of carcinoembryonic antigen-related cell adhesion molecule 1 (*CEACAM1*) were not changed in cytokine-sEVs (0.99 ± 0.70, *p* = 0.79) (Figure 5c).

### 3.6. Cytokine-sEV-Treated hCMEC/D3 Cells Show a Proinflammatory Profile

Previous studies have suggested that the molecular cargo carried within EVs can be transferred to the recipient cells [36]. Therefore, we evaluated the levels of *miRNA-155, miRNA-126-3p* and *miRNA-24* in naive BECs treated with cytokine-sEVs or quiescent sEVs by qPCR. These miRNAs have been found to be increased, decreased or unchanged in inflammation in the brain endothelium, respectively [37]. The levels of *miRNA-155-5p* were significantly upregulated in hCMEC/D3 cells after 24 h of incubation with cytokine-sEVs (1.46 ± 0.15-fold change, *p* = 0.0285), whereas the *miRNA-155-5p* levels were unchanged after treatment with quiescent sEVs (0.98 ± 0.19, *p* = 0.9728) (Figure 6a). The expression of *miRNA-126-3p* and *miRNA-24* was not altered following the sEV treatment (Figure 6a). 

In addition, we also investigated the effects of sEV treatment on the mRNA levels of genes that are involved in endothelial functions in hCMEC/D3 cells. *ICAM1* and *VCAM1* mRNA was upregulated after the treatment with cytokine-sEVs (1.55 ± 0.32, *p* = 0.0007 and 1.45 ± 0.13-fold change, *p* = 0.0053, respectively), whereas they were not changed after treatment with quiescent sEVs (Figure 6b). mRNA levels of the TJ protein claudin-5 (*CLDN5*) and occludin (*OCLN*) were not changed following EV incubation. Immunoblotting using primary antibodies against ICAM1, VCAM1 and GAPDH (Figure 6c) confirmed a significant upregulation of *ICAM1* and *VCAM1* in recipient BECs (166 ± 33% and 534 ± 50% change over untreated cells, *p* = 0.022 and *p* = 0.005, respectively) compared to untreated hCMEC/D3 cells (Figure 6d).

Finally, we investigated the role of VCAM1 and ICAM1 proteins in modulating cytokine-sEV-induced T-cell adhesion to the brain endothelium. Blocking ICAM1 did not prevent cytokine-sEV-induced T-cell adhesion when compared to IgG-treated hCMEC/D3 cells (1.93 ± 0.45 and 2.03 ± 0.14-fold change, respectively). By contrast, blocking VCAM1 significantly reduced T-cell adhesion to BECs (1.31 ± 0.27-fold change and *p* = 0.040). In addition, the simultaneous blockade of VCAM1 and ICAM1 completely prevented the effect of cytokine-sEV on T-cell binding to hCMEC/D3 cells (1.04 ± 0.34-fold change and *p* = 0.017) (Figure 6e). 

## 4. Discussion

The findings from the present study indicate that the treatment of BECs with cytokines promotes the release of sEVs with proinflammatory properties that can induce cerebrovascular dysfunction in naïve BEC cells by reducing the endothelial resistance and increasing leukocyte adhesion to the brain endothelium. Furthermore, the cytokine-induced release of sEVs carry a molecular material that is able to modulate the RNA and protein levels within the recipient endothelial cells that correspond with changes in the barrier and adhesion properties of the cells. These data suggest that BEC-derived sEVs may contribute to the progression of endothelial injury during neuroinflammation in the CNS (Figure 7). 

In accordance with most studies analysing the EV number after an inflammatory challenge [38], our results suggested that BECs shed more EVs after stimulation with TNFα and IFNy. Additionally, we found that the number of cytokine-sEVs was significantly higher than that of L-EVs after treatment with proinflammatory cytokines. This observation differs from Dozio et al., who reported that TNFα-stimulated hCMEC/D3 cells secreted equivalent amounts of sEV and L-sEVs [20]. Hypoxic lung epithelial cells were found to secrete a higher quantity of L-EV-protein than sEVs [39]. However, other reports have described an increased number of sEVs in comparison to L-EVs isolated from CCM under basal conditions [40,41]. Differences in the type and number of EVs released following inflammatory insult are likely to be specific of the model and treatment used and may also depend on the protein packaging within the EVs. Our results indicate that sEVs may play a more relevant role than L-EVs in modulating vascular functions during inflammation. However, the systematic comparison of L-EV and sEV functions in the brain endothelium is necessary to draw more robust conclusions about potential different roles of these two subsets of EVs in cerebrovascular functions.

We observed that the cytokine treatment had no effect on the size distribution (mean and mode) of sEVs and L-EVs, as has been previously reported [20]. However, while we found no changes in the mean and mode of sEVs compared to L-EVs, other studies have shown an increase in the mode and/or mean of L-EVs compared to sEVs, albeit with a great amount of overlap in the size distribution of the EVs [20,40,41]. This discrepancy is likely to occur due to differences in the EV isolation techniques, which have a different ability to recover pure EV populations [12]. Nevertheless, we found that the cytokine-L-EV size distribution was larger than the rest of the studied EVs evidenced by an increase in D90. Therefore, we are confident that our method of isolation was enriched for the different subsets, although a high heterogeneity was observed. 

The isolation of sEVs by differential ultracentrifugation has been reported to co-isolate soluble proteins [12]. For this reason, we measured the quantity of proinflammatory cytokines (TNFα and IFNy) remaining on the isolated sEVs. We found that the concentration of TNFα and IFNy in the sEV fraction was less than 1% of the original concentration to which the BECs were exposed. Moreover, this concentration was 100 times smaller than the lowest dose of TNFα and IFNy that induced an inflammatory response in the hCMEC/D3 cells [27]. Nevertheless, we cannot completely exclude that some of the responses of recipient BECs to cytokine-sEV may have been due to free TNFα and IFN, or to the possibility that some cytokines may also be carried within the sEVs [42,43]. 

Our results are in accordance with the general view that sEVs derived from proinflammatory cytokine-activated cells are capable of inducing cytokine-like effects in recipient cells [44]. However, most studies have investigated the role of EV communication and their impact on vascular functions using plasma- or serum-derived EVs with a multicellular origin [45]. To our knowledge, this is the first study to evaluate the role of EVs shed from the brain endothelium in modulating vascular functions during inflammation.

In the present study, we observed that cytokine-sEVs decreased the endothelial resistance, whereas quiescent sEVs have no effect on the TEER. These results are in accordance with previous reports that found that non-brain endothelial and other cellular origin EVs affect the endothelial resistance [18,46]. A decreased TEER is related to BBB permeability and has been associated with the migration of bloodborne molecules into the brain parenchyma [22]. Therefore, it is likely BEC-derived sEVs are involved in the extravasation of these molecules during inflammation at the BBB. We also showed a small transient effect of sEVs on the permeability of small tracers. Future experiments will help elucidate what other aspects of BBB permeability are affected by sEVs in inflammation.

We also observed that T-cell adhesion to BECs was twice as high in the presence of proinflammatory cytokines compared to cytokine-sEVs by themselves. Previous reports have reported a similar difference, although these differences vary according to the cell type and inflammatory stimulus [47,48]. We speculate that sEVs might finetune the response of naïve endothelial cells to neuroinflammation. A recent study showed that naïve HUVECS and human monocytic cells (THP-1) treated with EVs derived from human primary umbilical vein endothelial cells (HUVECs) stimulated with TNFα induced proinflammatory markers and increased monocyte adhesion and transmigration in vitro [48]. The data suggest that the secretion of inflammation-induced endothelial sEVs modulates the capability of naïve endothelial cells in binding leukocytes.

The modulation of cellular functions by sEVs is mediated by different mechanisms, including the release of EV cargo and/or protein–protein interaction [49]. Specifically, *miRNA-155-5p* has been previously shown to be transferred from sEVs to modulate the functions of the acceptor cells [50]. *MiRNA-155-5p* is a master regulator of cellular inflammation [51,52]. Previous research in our group demonstrated that endogenous *miRNA-155-5p* expression affected the cell activation and function of brain endothelial cells in inflammation [25,28]. Here, we observed an increased expression of *miRNA-155-5p* in both cytokine-sEVs and naïve BECs treated with cytokine-sEVs, while other inflammation-associated miRNAs were not changed. This was consistent with the current understanding that there is a variability between the expression of vesicular RNAs and the secreting cells/tissue [53]. However, whether *miRNA-155-5p-*enriched sEVs modulate TEER and/or leukocyte adhesion to BECs is still unknown. 

Only ICAM1 and VCAM1 protein levels were upregulated in recipient BECs after incubation with cytokine-sEVs, whereas no changes were found in occludin and claudin-5. This suggests that a cytokine-sEV-induced reduction in endothelial resistance is not due to a downregulation of TJ proteins but may be due to alterations of the subcellular location of proteins such as ZO-1 or VE-cadherin. This is supported by a study using plasma-derived EVs from MS patients, which showed that the decreased endothelial resistance was caused by reorganisation of the junctional complexes [18]. We also confirmed that blocking VCAM1 with neutralising antibodies [54] decreased the T-cell adhesion to BECs (Figure 6e). This result was not unexpected, since this protein has been described to play a crucial role in modulating leukocyte adhesion to the endothelium [33]. As the blocking of ICAM1 did not affect the endothelial functions, we speculate that the ICAM1 contribution to sEV-induced T-cell adhesion is minor in comparison to VCAM1, given that the ICAM1 protein levels in the recipient cells were lower than VCAM1 after cytokine-sEV treatment. 

Interestingly, Cerutti et al. showed that the overexpression of *miRNA-155-5p* in brain endothelial cells indirectly increased the levels of ICAM1 and VCAM1 [28]. Therefore, we speculate that the *miRNA-155*/ICAM1/VCAM1 axis is likely to be involved in inducing the effect of cytokine-sEVs in the brain endothelium. However, we cannot rule out the possibility of alternative pathways mediating an increase in VCAM1 and ICAM1.

## 5. Conclusions

The findings from the current study indicated a novel role for brain endothelial-derived sEVs in promoting and perpetuating endothelial dysfunction during inflammation. Our data suggest that this mechanism may be driven in part by the transmission and/or induction of the proinflammatory modulator *miRNA-155-5p*, as well as adhesion molecules ICAM1 and VCAM1. The results from this manuscript propose a novel mechanism of communication among brain endothelial cells during neuroinflammation.

## Figures and Tables

**Figure 1 pharmaceutics-13-01525-f001:**
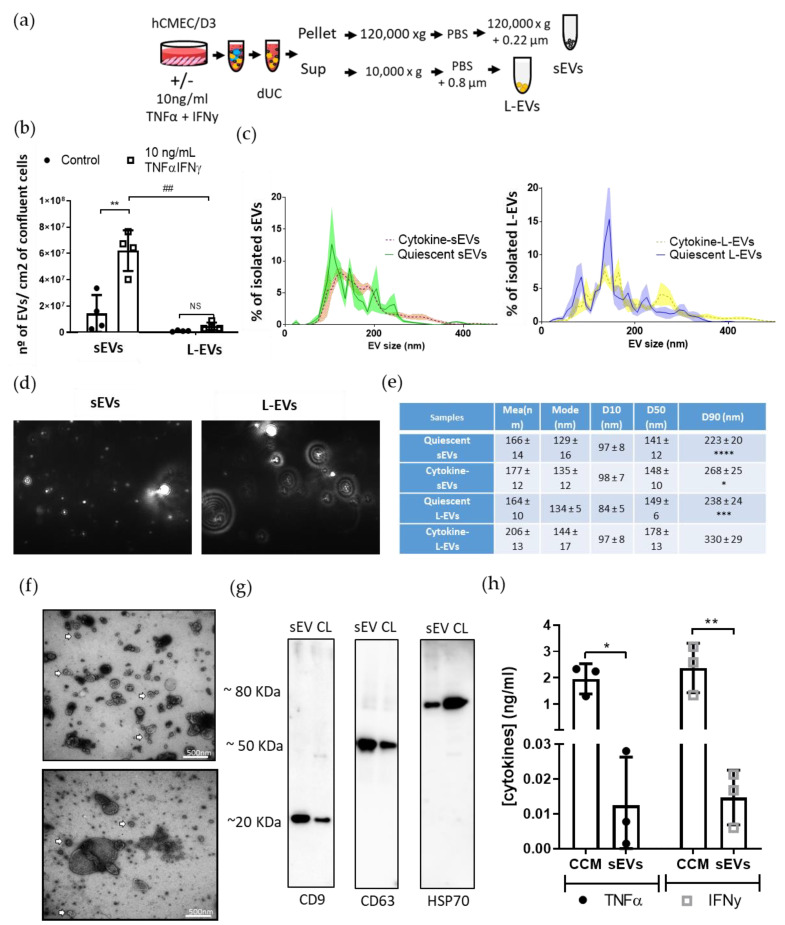
Characterisation of extracellular vesicles in inflammation. (**a**) Graphic diagram of the isolation of extracellular vesicles (EVs) both small (sEVs) and large (L-EVs) from untreated hCMEC/D3 cells (quiescent sEVs or L-EVs) and those incubated with 10-ng/mL TNFα and IFNy (cytokine-sEVs or -L-EVs) for 24 h using a combination of ultracentrifugation (dUC) and filtration steps. (**b**) Nanoparticle tracking analysis (NTA) measurements of isolated EVs revealed significantly increased numbers of cytokine-sEVs after the cytokine treatment compared to the controls. (**c**) Size distribution measured by NTA showed that most isolated EVs were smaller than 200 nm in all EV subsets (left image shows sEVs, whereas the right image shows L-EVs). (**d**) Representative images of cytokine-sEVs and L-EVs from the Nanoparticle Tracking Analysis (**e**) The table summarises the mean; mode and D10, D50 and D90 diameter sizes of sEVs and L-EVS. (**f**). Transmission electron microscopy (TEM) images of cytokine-sEVs and cytokine-L-EVs. Arrows point to sEVs in both images. (**g**) Immunoblots for CD9, CD63 and HSP70 in cytokine-sEVs (sEV) and cell lysate (CL) controls. (**h**) Concentrations of TNFα and IFNy in the cytokine-sEV fractions and cell-conditioned medium (CCM). Data are shown as the mean ± SEM, *n* = 2 (**d**), *n* = 3 (**f**) and *n* = 4 (**b**,**c**). The means were compared with one-way ANOVA, followed by Tukey’s post-hoc test. * *p <* 0.05, ^##^ or ** *p <* 0.01, *** *p <* 0.001 and **** *p* < 0.0001.

**Figure 2 pharmaceutics-13-01525-f002:**
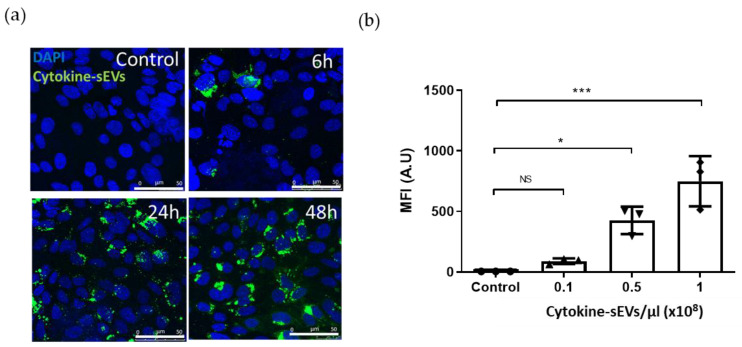
Characterisation of the uptake of cytokine-sEVs by naïve brain endothelium. (**a**) Confocal images of the uptake of 0.5 × 10^8^ DiO-labelled cytokine-sEVs/µL by human microvascular brain endothelial cells (hCMEC/D3 cells) at different time points (0, 6, 24 and 48 h). (**b**) Flow cytometry measurements of DIO-labelled cytokine-sEVs incubated with hCMEC/D3. The graph shows a quantification of the median fluorescent intensity (MFI) of the uptake of cytokine-sEVs (0.1, 0.5 and 1×10^8^ sEVs/µL) by naïve hCMEC/D3 cells. Data is shown as the mean ± SEM (*n* = 3). The means were compared with one-way ANOVA followed by Tukey’s multiple comparison test. ** p* < 0.05 and **** p* < 0.001.

**Figure 3 pharmaceutics-13-01525-f003:**
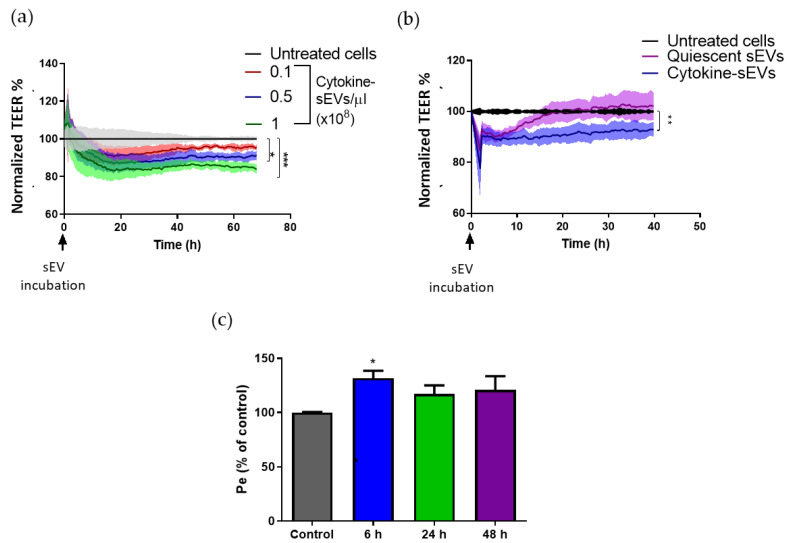
Cytokine-sEVs decrease the transendothelial resistance of hCMEC/D3 cells. (**a**) TEER values following hCMEC/D3 cells with varying concentrations of cytokine-derived small EVs (cytokine-sEVs) (0.1, 0.5 and 1 × 10^8^ sEVs/µL). Data are shown as the percentage fold change of treated relative to untreated cells at each time point. (**b**) Comparison of the effects of 0.5 × 10^8^ sEVs/µL cytokine-sEVs and quiescent sEVs on the TEER values. (**c**) Para-cellular permeability experiment of 70-KDa FITC-Dextran after a 0.1 × 10^8^ cytokine-sEVs/ul stimulation for 6, 24 and 48 h. Data is shown as the mean ± SEM. The means are compared by two-way ANOVA followed by Tukey’s post-hoc for multiple comparisons, *n* = 3 (**b**) and 4 (a). * *p* < 0.05, ** *p* < 0.01, *** and *p* < 0.001 compared to untreated cells.

**Figure 4 pharmaceutics-13-01525-f004:**
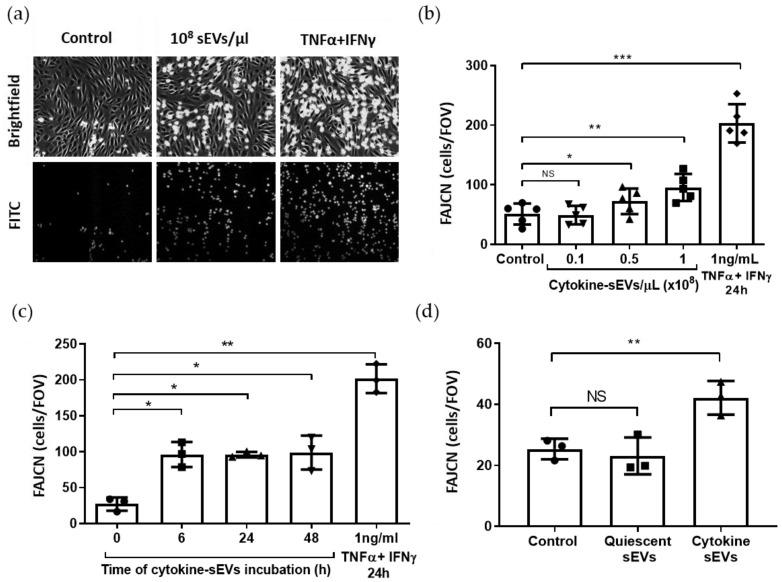
Cytokine-sEVs increase Jurkat T-cell adhesion to the brain endothelium. (**a**) Representative brightfield (top) and fluorescent (FITC, bottom) images of Jurkat T-cell adhesion (round-shaped cells) to hCMEC/D3 cells (spindle-shaped cells) under flow conditions in their absence (control, left panels), with the addition of cytokine-derived small EVs (cytokine-sEVs) (middle panels) or cytokines (1-ng/mL TNFα + IFNy, right panels). (**b**) Quantification of the number of firmly adhered T cells per field of view (FAJCN) (FOV 640 × 480 μm) following incubation with increasing doses of cytokine-sEVs (0.1, 0.5 and 1 × 10^8^ cytokine-sEVs/µL) for 24 h. (**c**) Time–course analysis (6, 24 and 48 h) of the effect of 1 × 10^8^ cytokine-sEVs on FAJCN on naïve endothelium. (**d**) Comparison of cytokine-sEVs and quiescent sEVs (0.5 × 10^8^ sEVs/µL for 24 h) on FACJN in naïve hCMEC/D3 cells. Data is shown as the mean ± SEM, and *n* = 4 (**b**,**c**) and 3 (**d**). The means are compared by one-way ANOVA followed by Dunnett’s post-hoc for multiple comparisons, * *p* < 0.05 and ** *p* < 0.01 and *** *p* < 0.001 to the untreated condition.

**Figure 5 pharmaceutics-13-01525-f005:**
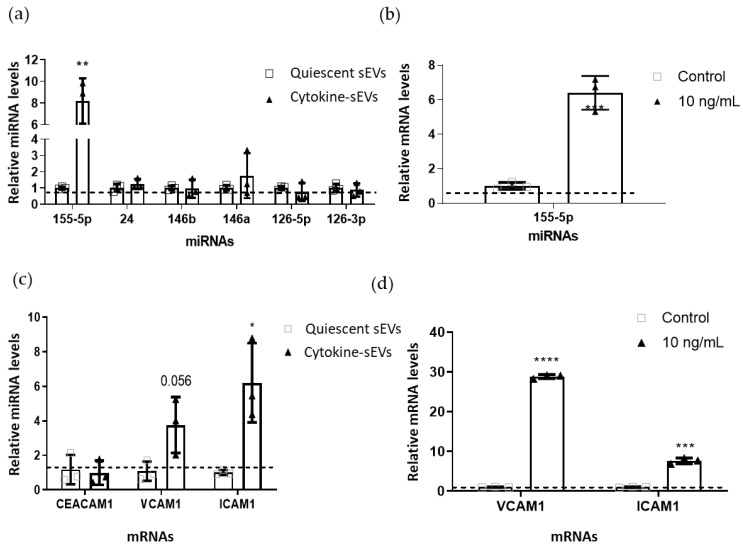
Characterisation of inflammation-related miRNAs and mRNAs in sEV and secreting hCMEC/D3 cells. (**a**) Quantification of levels of *miRNA-155-5p, miRNA-146a, miRNA-146b, miRNA-126-5p, miRNA-126-3p* and miRNA-24 in cytokine-sEVs and quiescent sEVs. (**b**) Levels of *miRNA-155-5p* in naïve hCMEC/D3 cells and those treated with 10-ng/mL TNFα and IFNy. (**c**). mRNA levels of vascular cell adhesion molecule (*VCAM1*), intercellular adhesion molecule (*ICAM1*) and carcinoembryonic antigen-related cell adhesion molecule 1 (*CEACAM1*) in cytokine-sEVs and quiescent sEVs. (**d**) Levels of *VCAM1* and *ICAM1* mRNA in naïve hCMEC/D3 cells and those treated with 10-ng/mL TNFα and IFNy. Data are shown as the mean ± SEM (*n* = 3). The means were compared by unpaired two-tailed *t*-tests, * *p <* 0.05, ** *p <* 0.01, *** *p <* 0.001, **** *p <* 0.0001 compared to quiescent sEV cargos.

**Figure 6 pharmaceutics-13-01525-f006:**
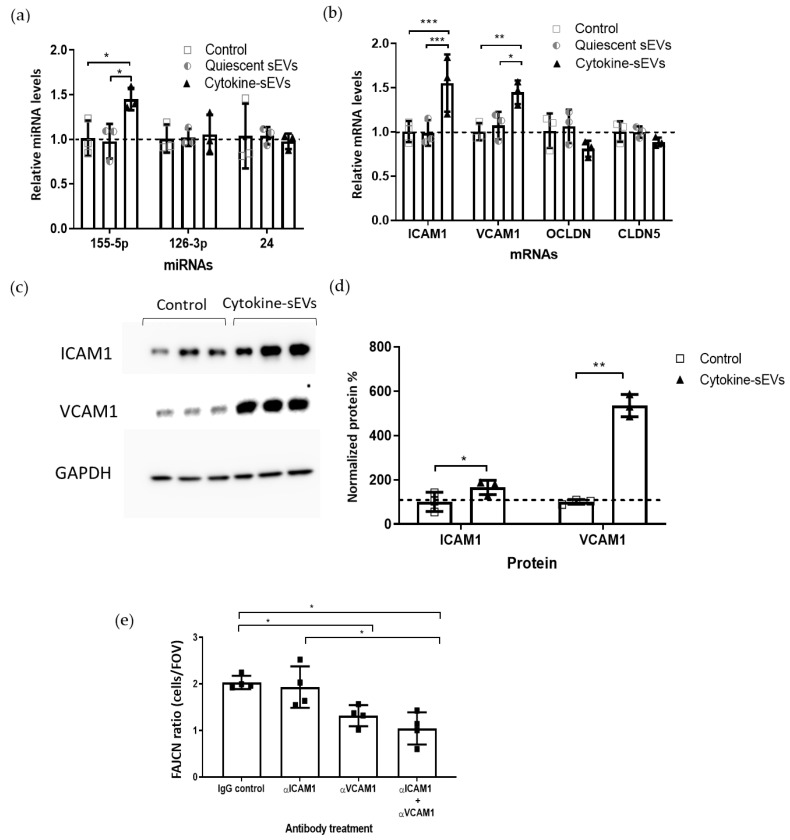
Cytokine-sEVs induce a proinflammatory profile in recipient hCMEC/D3 cells. (**a**) Relative levels of *miRNA-155-5p*, *miRNA-126-3p* and *miRNA-24* in hCMEC/D3 cells after treatment with cytokine- or quiescent sEVs. (**b**) Relative levels of *ICAM1, VCAM1,* occludin (*OCLDN*) and claudin-5 (*CLDN5*) on hCMEC/D3 cells treated with cytokine- or quiescent sEVs for 24 h. (**c**,**d**) Immunoblot and quantification of cytokine-sEV-treated or untreated (control) hCMEC/D3 cells for ICAM1 and VCAM1. (**e**) Effect of blocking of ICAM and VCAM alone and in combination with the adhesion of Jurkat T cells to untreated or cytokine-sEV-treated hCMEC/D3 cells. Data is shown as the mean ± SEM, *n* = 3 (**a**,**b**,**d**) and *n* = 4 (**e**). The means were compared by two-way ANOVA followed by Tukey’s comparison test (**a**,**b**) or paired two-tailed *t*-test (**d**), * *p <* 0.05, ** *p* < 0.01 and *** *p <* 0.001 in comparison to untreated cell levels (**a**,**b**,**d**) or to the IgG control (**e**).

**Figure 7 pharmaceutics-13-01525-f007:**
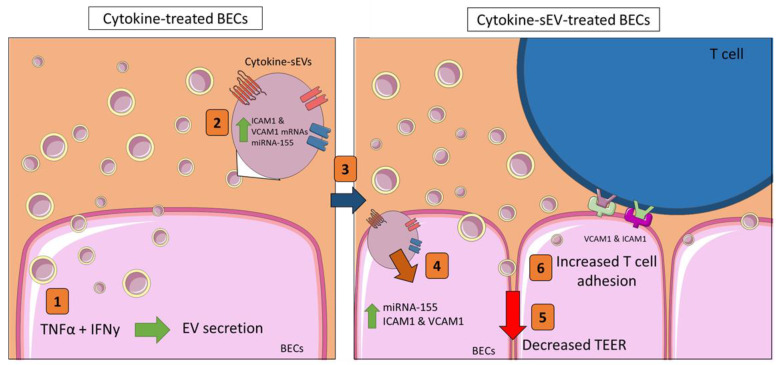
The proposed model for the cytokine-sEV role in modulating cerebrovascular functions. (1) The incubation of hCMEC/D3 cells with 10 ng/mL of TNFα and IFNy increased the secretion of small EVs (cytokine-sEVs). (2) These cytokine-sEVs carried elevated levels of the proinflammatory *miRNA-155*, as well as mRNAs of *VCAM1* and *ICAM1*. (3) Naïve hCMEC/D3 cells take up cytokine-sEVs, leading to increased intracellular levels of *miRNA-155* and ICAM1 and VCAM1 in recipient hCMEC/D3 cells (4). Consequently, the endothelial resistance is decreased (5), while the firm adhesion of Jurkat T cells (6) is increased in recipient endothelial cells.

## Data Availability

The datasets used and/or analysed during the current study are available from the corresponding author upon reasonable request.

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
