# Peer review of "Endothelial-Derived Extracellular Vesicles Induce Cerebrovascular Dysfunction in Inflammation"

_pharmaceutics, 2021, doi:10.3390/pharmaceutics13091525_

Round 1

Reviewer 1 Report

The authors proposed an original autocrine regulation on Blood-brain barrier endothelial cells mediated by extracellular vesicles.

The manuscript will be enriched by adding some additional experiments/data:

  • TEER values are the unique "control" to deal with a BBB phenotype. Please provide reviewers permeability experiments with Dextran/small molecules such as Lucifer yellow, sodium fluorescein or something convenient with a BBB permeability.
  • TEM pictutes showed sets but melted in a cloud of protein aggregates, debris and other EVs. Can you be sure that the sEVs you used are de facto sEVs ? Please provide NTS or related techniques data to prove it. Moreover, the protein characterization by western blot is quite light to confirm the sEV main proteins.
  • What is going on with the addition of exogeneous mi-155p ?
  • Did the authors have some data from T-cell transmigration after cytokines-sEVs treatment ?

Author Response

  • TEER values are the unique "control" to deal with a BBB phenotype. Please provide reviewers permeability experiments with Dextran/small molecules such as Lucifer yellow, sodium fluorescein or something convenient with a BBB permeability.

We appreciate the reviewer’s concern on showing only TEER values. We have added data of an experiment evaluating the permeability of 70 KDa FITC-Dextran after the incubation with 0.1x108 cytokine-sEVs/µl  at 6, 24 and 48h(Figure 3C). Here we showed a significant but transient effect on para-cellular permeability after 6h of incubation with cytokine-sEVs.

  • TEM pictutes showed sets but melted in a cloud of protein aggregates, debris and other EVs. Can you be sure that the sEVs you used are de facto sEVs ? Please provide NTS or related techniques data to prove it. Moreover, the protein characterization by western blot is quite light to confirm the sEV main proteins.

We thank the reviewer from bringing up this point. The “cloud” showed in the picture is likely to be artefacts caused by the staining with uranyl acetate. As a pre-caution method, residual levels of TNFα and IFNγ were measured (Figure 1H). Nanoparticle tracking analysis images have been added to figure 1D as requested.

Regarding the EV protein characterization by western blot, MISEV guidelines from 2018 (PMID: 30637094) recommended to use at least three EV markers. Therefore, we think the three markers used in this work are appropriate for the characterization of sEVs.

  • What is going on with the addition of exogeneous mi-155p ?

Investigating the exogenous effect of miRNA-155 is technically challenging. The most accurate approach would be to generate a stable transfected brain endothelial cell line where miRNA-155 had been silenced. However, this experiment was beyond the possibilities of the project. As explained in the discussion section, based on the literature and what it is known of miRNA-155 in BBB function, it is likely that sEV-mediated miRNA-155 increase will impact on T cell adhesion to brain endothelial cells. Future experiments will evaluate the precise mechanism by which this miRNA-155-5p-enriched sEVs subpopulation mediates T cell adhesion and how blocking of miRNA-155-5p may affect the observed effects.

  • Did the authors have some data from T-cell transmigration after cytokines-sEVs treatment ?

Understanding the role of sEVs would be very interesting and informative for the characterization of sEVs at the BBB. However, no migration data was generated for this project.

Reviewer 2 Report

The authors investigated the role of brain endothelial-derived sEVs in cerebrovascular dysfunction during inflammation and indicate a new communication mechanism between endothelial cells in neuroinflammatory diseases. This mechanism is based on the involvement of the proinflammatory modulator miRNA-155-5p and the adhesion molecules ICAM1 and VCAM1. The work is complete and well-structured and the results well described. The discussion is clear and sufficiently supported by the literature. Figure 7 well summarizes the proposed mechanism. The evaluations and data reported by the authors are interesting and stimulating for subsequent studies. I think that the studies aimed at a better understanding of the mechanisms involved in the development of neurological diseases have an important scientific significance given the high worldwide incidence rate of these diseases. Therefore, I believe that the mechanism identified in this study can make a significant contribution in the search for an effective therapy.

Minor review:

Abbreviations such as BBB, EVs, and BEC already present in the abstract should be reported as such in the following sections without further repetition.

page 13 line 387: replace miRNA-126-5p with miRNA-126-3p shown in Figure 6A.

Different abbreviation of occludin in the text and in figure 6B.

Author Response

We thank the reviewer for the feedback provided. We can confirm that minor comments have been amended in the main text

Round 2

Reviewer 1 Report

I would like to thank the authors to have answered all my questions and done all the modifications required to make this paper acceptable for publication.